# Pulse Proteins: Processing, Nutrition, and Functionality in Foods

**DOI:** 10.3390/foods14071151

**Published:** 2025-03-26

**Authors:** Valeria Messina, Daniel J. Skylas, Thomas H. Roberts, Peter Valtchev, Chris Whiteway, Ziqi Li, Andreas Hopf, Fariba Dehghani, Ken J. Quail, Brent N. Kaiser

**Affiliations:** 1School of Life and Environmental Sciences, Faculty of Science, University of Sydney, Camperdown, NSW 2006, Australia; thomas.roberts@sydney.edu.au (T.H.R.); peter.valtchev@sydney.edu.au (P.V.); brent.kaiser@sydney.edu.au (B.N.K.); 2Australian Export Grains Innovation Centre, North Ryde, NSW 1670, Australia; chris.whiteway@aegic.org.au (C.W.); ken.quail@aegic.org.au (K.J.Q.); 3School of Chemical and Biomolecular Engineering, Faculty of Engineering, The University of Sydney, Sydney, NSW 2006, Australia; zili5377@uni.sydney.edu.au (Z.L.); ahop9012@uni.sydeny.edu.au (A.H.);; 4Sydney Institute of Agriculture, The University of Sydney, 380 Werombi Road, Brownlow Hill, NSW 2570, Australia

**Keywords:** pulses, protein ingredients, functionality, food processing

## Abstract

Pulses are grown worldwide and provide agronomic benefits that contribute to the sustainability of cropping systems. Pulses are high in protein and provide a good source of carbohydrates, dietary fibre, vitamins, minerals, and bioactive constituents. Crops such as lupins, chickpeas, faba beans, field peas, lentils, and mung beans, and the diversity of varieties among them, provide enormous opportunities for processing protein ingredients for use in new and existing food formulations. This review highlights the nutritional properties of pulses, protein quality, functionality, and applications for pulse protein ingredients. Understanding the functionality of pulse proteins, and the unique properties between different pulses in terms of solubility, water- and oil-holding capacity, emulsification, gelation, and foaming properties, will help maximise their use in plant-based meat and dairy alternatives, beverages, bakery products, noodles, pasta, and nutritional supplements. In this review, researchers, food technologists, and food manufacturers are provided with a comprehensive resource on pulses, and the diverse applications for pulse protein ingredients within the context of food manufacturing and the constantly evolving food technology landscape.

## 1. Introduction

The rapid rise in the global population, which is expected to increase by nearly 2 billion people over the next thirty years, as well as the depletion of natural resources, has led to growing interest in the exploitation of plant-based proteins as an alternative to animal-derived foods [1]. The natural diversity and abundance of plant-based proteins provide an alternative and realistic pathway for reducing animal product consumption. However, the formulation of plant-based proteins into new and existing foods must meet consumer preferences for taste, price, and convenience [2]. Plant-based proteins offer a more sustainable food source compared to animal-derived products, as they produce lower greenhouse gas emissions, and require less water and land [3]. Other significant drivers behind the increasing demand for plant-based proteins include health benefits, animal welfare concerns, and changing dietary patterns, particularly the rise in flexitarian diets, as well as vegetarian and semi-vegetarian diets, with the latter being primarily vegetarian with the occasional inclusion of meat or fish [4].

Legumes belong to the Fabaceae family (Leguminosae) and represent one of the largest groups of angiosperms (flowering plants). According to the Food and Agriculture Organization (FAO) and the Codex Alimentarius Commission, pulses are defined as the dry and edible seeds of legumes with low-fat content and, therefore, distinguished from leguminous oil seeds, like soybeans and groundnuts [5].

Pulses are increasingly being explored as a nutritious, economical, and sustainable source of plant-based proteins for human consumption and animal feed. Pulses are high in protein, generally twice as high as cereals, ranging from 15 to 30% (dry basis, db.), and provide a good source of carbohydrates, dietary fibres, vitamins, minerals, and bioactive constituents [6,7,8]. Pulse proteins have unique functional properties, including water- and oil-binding capacity, solubility, emulsification, foaming, and gelation, which can be exploited in the development of new and existing foods [9,10]. Furthermore, the consumption of pulses in the human diet is associated with protective effects against various chronic diseases, including obesity [7,11], type 2 diabetes [12], cardiovascular disease [13,14], and cancer [15,16].

The rationale for this review is that there have been major advances in the production and processing of pulse proteins in recent years, which require critical summary and evaluation. The aim of the review is to provide a comprehensive resource on pulses, encompassing nutritional composition, protein extraction or fractionation methods, functional properties, and potential applications of pulse protein ingredients in foods and future trends. Furthermore, we also highlight the challenges and limitations of pulses and pulse protein ingredients, so future research can focus on overcoming these barriers and unlock their full potential in more sustainable food production systems.

## 2. Worldwide Pulse Production

The major types of pulses grown worldwide include dry beans, chickpeas, peas, cowpeas, broad beans and horse beans, lentils, pigeon peas, and lupins [17]. Worldwide pulse production from 2018 to 2022 is reported in Table 1 [18], with the largest category being dry beans, with an annual production of around 27 million metric tonnes (mmt). Dry beans include several crops, such as the common bean, mung bean, black gram, and adzuki bean. According to the OECD-FAO Agricultural Outlook (2022–2031), worldwide pulse production is forecast to increase from around 103 mmt (2022) to over 120 mmt by 2031 [19].

Pulses are traded in many different export markets, with some markets requiring stricter processing and quality requirements than others. Figure 1 summarises the various stages of pulse production, processing, and value-added opportunities to maximise the use of pulses and their ingredients.

## 3. Composition of Pulses

The proximate composition of pulses varies significantly depending on factors such as species, growing conditions, and geographical location. A summary of the proximate composition of pulses grown around the world is provided in Table 2 [20,21,22,23,24,25,26,27,28,29,30,31,32,33,34,35,36,37,38,39,40,41,42,43,44].

### 3.1. Protein

The protein content of whole-seed pulses, containing the hulls, generally ranges from 20 to 30% (db.), depending on the type of pulse [45]. For dehulled grains of varieties of two lupin species, *Lupinus angustifolius* (Jurien) and *Lupinus albus* (Murringo), the protein content was found to be 41.0% and 43.1% (db.), respectively [18]. Similar results were reported for other dehulled lupin varieties, with protein contents ranging from 41.5% to 46.2% (*L. angustifolius*) and 46.5–48.2% (*L. albus*) [46]. The protein content of whole-seed material from 10 different cultivars of faba bean (*Vicia faba*) varieties ranged from 269.5 to 295.3 g kg^−1^ (db.) [47]. The faba bean variety PBA Rana had the highest protein content across both sites and seasons, ranging from 287.6 to 295.3 g kg^−1^ (db.). Therefore, PBA Rana might be a suitable variety for processing protein ingredients on a commercial scale. Dehulled faba bean has been found to have a protein content of 33.0 g/100 g (db.) [36], while another study reported similar protein contents of 275–324 g kg^−1^ (db.) for 11 faba bean genotypes, grown in three locations in western Canada, over two consecutive seasons (2006 and 2007) [48]. In this second study, protein contents of 242–275 g kg^−1^ (db.) were reported for 10 field pea (*Pisum sativum*) genotypes, grown in four locations in Saskatchewan, over the same seasons. The protein content of green and red lentil (*Lens culinaris*) varieties was found to be 32.6% (Matilda) and 30.3% (Digger), respectively [29].

Whole-seed material of three mung bean (*Vigna radiata*) varieties (Crystal, Satin II, and Celera II-AU), grown in different regions across Queensland (Warra and Hermitage) and New South Wales (Liverpool plains and Northern NSW), had protein contents ranging from 23.6 to 30.1% (db.) [49]. Mung bean varieties grown in the Warra region had the highest protein content compared to the other regions, highlighting that environmental conditions can impact protein content. The proportion of essential AAs relative to total AAs was highly conserved between varieties and regions (38.1–38.7%). Limiting AAs for mung bean varieties were cysteine, methionine, and in some cases, tryptophan. Sulfur-containing AAs (methionine and cysteine) are generally deficient in pulses, with higher levels found in cereals, whereas pulses are higher in lysine compared to cereals. This shows the complementary nature of amino acid profiles between pulses and cereals, whereby blending these two different plant sources is a simple and effective means of improving protein quality [50].

The major types of proteins found in pulses are albumins (soluble in water) and globulins (soluble in dilute salt solutions) [51]. Albumins generally contribute less (10–20%) than globulins to total seed protein and have molecular weights ranging from 5 to 80 kDa but contain higher levels of cysteine compared to globulins. Globulins constitute about 70–80% of the total seed proteins and function primarily as storage proteins. Legumin (11S) and vicilin (7S) are the major types of globulins, with 11S legumins containing acidic and basic subunits with molecular weights of 40 and 20 kDa, respectively [52]. Vicilins have molecular weights ranging from 175 to 180 kDa and are the main storage proteins in mung bean, kidney bean (*Phaseolus vulgaris*), and cowpea (*Vigna unguiculata*), accounting for up to 88% of the total globulin content [53].

By definition, albumins have a higher solubility than globulins and are capable of interacting and competing with starch for water. For proteins from lentils, higher digestibility was shown in globulins compared to albumins because of the lower content of cysteine and a smaller number of disulfide bonds. An increase in protein digestibility for both globulins and albumins observed in the presence of starch may be related to the opening of compact protein structures due to their binding to the starch granule surface, as well as the formation of new bonds, increasing accessibility to proteolytic enzymes [54].

### 3.2. Carbohydrates

Carbohydrates generally contribute 50% to 65% of the seed weight and are classified into three major groups based on their chemical structure, namely, simple sugars, oligosaccharides, and polysaccharides, with the latter referred to as complex carbohydrates [55]. Starch is the major carbohydrate component (22–45%) and acts as an energy reserve for seed germination. Lupin seeds contain negligible starch but instead have higher levels of fat and dietary fibre compared to other pulses. Starch, comprising amylose and amylopectin, is deposited in granules, which are arranged in a partially crystalline structure. Pulse starches contain higher amylose content (>30%) compared to cereal starches, leading to a lower glycaemic index, and higher levels of resistant starch, which could be of potential use in food formulations for diabetics [56].

A range of carbohydrate contents for kabuli chickpea (*Cicer Arietinum*) varieties have been reported: 52.7% (Kimberley Large), 44.8% (Genesis 090), 40.5% (Genesis Kalkee), 44.4% (PBA Royal), and 44.5% (PBA Monarch) [26]. The corresponding starch contents for these varieties were 48.9%, 32.7%, 38.8%, 36.6%, and 38.9%, respectively. The physicochemical and technological properties of carbohydrates, especially starch, can vary among different pulses and their cultivars. Pastes made from pulse starches have a high tendency to retrograde and are more resistant to being ruptured during cooking compared to cereal starches. This improvement in the stability of both heat and mechanical properties makes them highly functional ingredients in many food applications. The total carbohydrate content of pulses (54.7–63.9 g/100 g) is lower in comparison to cereals, reportedly being up to 72 g/100 g in wheat [34].

Plant cell walls contain a complex mixture of polysaccharides (oligosaccharides, hemicelluloses, cellulose, gums, and pectins), lignins, and waxes, collectively referred to as non-available carbohydrate, which contributes the dietary fibre component [57]. They also have high fibre content (15–32%), comprising around 75% of insoluble fibre and 25% of soluble fibre [58]. Insoluble dietary fibre is beneficial for promoting laxation, while soluble dietary fibre helps reduce cholesterol levels and maintain post-prandial glucose levels. Both insoluble and soluble fibre can act as prebiotics, providing nutrients to gut microorganisms. Fibre-rich fractions from pulses can be incorporated into processed foods to enhance their fibre content. Despite their nutritional and health-promoting benefits, pulse fibres can also be utilised to improve the textural properties of food by binding and retaining fat and moisture [59,60,61].

### 3.3. Bioactive Compounds

Bioactive compounds in pulses play an important role in human nutrition, as diets rich in pulses are associated with a lower risk of several diseases, which is attributed to several non-nutritive bioactive and health-promoting compounds. Bioactive compounds can act as natural antioxidants and protect DNA from damage [62].

Phenolic compounds can chelate metals, inhibit lipid peroxidation, and scavenge free radicals. The major phenolic compounds in pulses are tannins, phenolic acids, and flavonoids. Genotypic variations can influence the phenolic and flavonoid biosynthetic pathways, and phenolic content can vary significantly among varieties [63]. Both phenolic content and antioxidant activities can also be affected by different processing methods. Roasting faba beans for 10, 20, 30, 60, and 120 min was shown to gradually decrease the total phenolic, flavonoid, and proanthocyanidin contents by 42%, 42%, and 30%, respectively [64]. The 2,2-diphenyl-1-picrylhydrazyl radical scavenging activity, total equivalent antioxidant capacity, and ferric-reducing antioxidant power were reduced by 48%, 15%, and 8%, respectively. When faba bean varieties with differing seed coat colours, including red (Rossa), green (Icarus), buff (Doza and Nura), and a white-coloured low-tannin breeding line, were either soaked, boiled, or autoclaved, subsequent losses in phenolic contents and antioxidant activities were observed due to leaching bioactive compounds into the soaking and cooking medium. Amongst the faba bean varieties, the white-coloured low-tannin breeding line had the lowest phenolic contents and antioxidant capacities [65].

The antioxidative contents of 10 faba bean varieties grown in field trials in South Australia over consecutive seasons (2016 and 2017) have been profiled [66]. The mean ferric-reducing antioxidant potential of the varieties ranged from 237 to 531 mg trolox equivalents 100 g^−1^; the total phenolics ranged from 258 to 571 mg gallic acid equivalents 100 g^−1^; and the total monomeric anthocyanins ranged from 12.7 to 21.0 mg cyanidin-3-glucoside equivalents 100 g^−1^. Significant variations in anthocyanin, phenolic, and antioxidant contents were observed between varieties, and the PBA Rana variety had the highest antioxidant and total phenolic contents. The phenolic acid and flavonoid composition of these same varieties were subsequently reported [67]. The most abundant flavonoid was catechin, whilst syringic acid was the phenolic acid found in high concentrations (72.4–122.5 mg/kg).

Saponins are known to have hypocholesterolemic, anticarcinogenic, and immune-stimulatory properties. Chickpeas constitute major sources of saponins in the human diet, followed by lupins, lentils, beans, and peas [68]. Higher levels of saponins were reported for Sweet Lupin (*L. angustifolius*) when compared to lentils, chickpeas, beans, and peas [69].

### 3.4. Off-Flavour Compounds

Off-flavours occurring in pulses vary depending on cultivar, seasonal variations, growing location, as well as harvesting, processing, and storage conditions. They are mainly related to aldehydes, alcohols, ketones, acids, pyrazines, and sulfur-containing compounds, which contribute to sensory perceptions. Off-flavour compounds are correlated with the presence of saponins, phenolic compounds, and alkaloids [70]. Off-flavours can develop during harvesting, processing, and storage and are related to enzymatic (e.g., lipoxygenase) or non-enzymatic reactions, effects of heat on sugars and amino acids (Maillard reactions), thermal degradation of phenolic acids, oxidative and thermal degradation of carotenoids, thermal degradation of thiamine, or potential contamination following solvent extraction [71].

Off-flavour in peas is mainly due to compounds such as *n*-hexanal, 3-*cis*-hexenal, *n*-pentyl furan, 2(1-pentenyl) furan, and ethyl vinyl ketone, which are lipoxygenase-derived contributors to grassy, beany, and green flavours. The bitterness of peas has been related to their saponin content and is also dependent on the pea cultivar [72]. Lupins mainly contain off-flavours, which are green/milky (*n*-pentanal), sperm (1-pyrroline), grassy, and sulfurous (dimethyl trisulfide [73]. Bitterness and astringency in lentils are related to catechin gallate and various forms of kaempferol glycosides and phenolic compounds [74]. Phenolic compounds in chickpeas are also responsible for bitterness and astringency [75]. Iso-flavones such as formononetin, biochanin A daidzein, and genistein are also responsible for bitterness [76]. Faba beans, which contain aromatic hydrocarbons, aldehydes, alkanes, alkenes, alcohols, ketones, furans, and other compounds such as tannins, contribute to off-flavours [77].

Off-flavours can be removed or modified using various technologies or processes, such as enzymatic treatment, soaking, thermal treatment, germination, solvent extraction, and fermentation. The removal of off-flavours is essential for the processing of food products or ingredients with acceptable palatability. One way to achieve this is to ensure that the development of off-flavours remains as low as possible during harvesting, processing, and storage. The flavour aspects of pulses and pulse-derived ingredients have been reviewed [70]. Figure 2 summarises the different processing techniques that can be applied to remove or modify off-flavours.

### 3.5. Antinutritional Factors

Antinutritional factors have a wide range of biological functions; their presence in pulses is part of an adaptation mechanism to protect them from adverse environmental conditions as well as providing a defence mechanism against insects. They can negatively affect the nutritional quality and bioavailability of nutrients but also exhibit health-benefiting properties. The antinutritional content of pulses depends on a range of factors such as genotype, seed maturity, seed tissue, environment, growing region, and seasonal variation [78]. Antinutritional factors include phytic acid, enzyme inhibitors (trypsin, chymotrypsin, and α-amylase inhibitors), lectins, tannins, saponins, and many other compounds, which have previously been reviewed [6]. Protease inhibitors of the serine proteases, trypsin, and chymotrypsin are also common in pulses [79]. Lectins or haemagglutinins found in pulses can inhibit the growth of animals by reducing the digestibility and biological value of dietary proteins [80]. A study showed that a suitable ratio of antinutrients to nutrients can reduce the negative effects of antinutrients on digestibility while playing a significant role in cellular processes such as anti-inflammatory and antioxidant activity [81].

Faba beans contain the pyrimidine glucosides, vicine, and convicine, which are present in the cotyledons of most varieties and are involved in pathogen defence mechanisms [82]. Vicine, convicine, and total vicine and convicine, reported for 10 commercial faba bean varieties, were in the ranges 4.5–7.4, 1.7–3.2, and 6.4–9.6 mg/g, respectively [47]. Ingestion of faba beans by individuals carrying a deficiency in the red blood cell enzyme glucose-6-phosphate dehydrogenase can lead to haemolytic anaemia, also known as favism.

Phytic acid or phytate is the main storage form of phosphorus in the seed and is found in all pulses but can be higher in some pulses compared to others. Phytic acid is a chelating agent that binds divalent cations, such as iron, zinc, and calcium, reducing their bioavailability.

Phytic acid plays a crucial role in plant metabolism, stress tolerance, and resistance to pathogens. Moreover, it may provide secondary health benefits in human diets, potentially lowering the risk of heart disease and diabetes [12]. It can also form complexes with proteins, through the formation of phytate–protein or phytate–protein–mineral complexes [83], which reduces the digestibility of proteins and bioavailability of amino acids.

A study on the effects of spontaneous fermentation and germination on the levels of antinutrients and iron in soybean, kidney bean, and mung bean revealed that these processing techniques significantly decreased antinutrient levels and increased iron content [84]. Several researchers have reported that germination, fermentation, and soaking (24 h) effectively reduced phytate content in maize and legumes, thereby enhancing the bioavailability of minerals such as iron and zinc [85,86].

The effects of popping, soaking, boiling, and roasting processes on antinutritional factors in chickpeas and red kidney beans, after soaking (24 h) and boiling, were found not to be significantly different to raw seeds when compared to popping and roasting, which significantly reduced phytic acid content in chickpeas (6–22%) and red kidney beans (16–39%) [87].

Dry fractionation can also have an impact on the phytic acid content of air-classified coarse and fine fractions. The phytate content in original flours of red lentil, yellow lentil, green pea, and kabuli chickpea has been reported to be 6.43, 7.39, 8.70, and 6.78 mg/g (db.), respectively [88]. The phytate content for all these pulses was lower in the air-classified coarse fraction and higher in the fine fraction, with the highest values being for yellow lentil (14.06 mg/g) and green pea (13.95 mg/g).

Members of the raffinose family of oligosaccharides (RFOs), consisting of raffinose, verbascose, and stachyose, are associated with digestive discomfort, causing gas production and flatulence, due to the anaerobic fermentation of these carbohydrates in the large intestine [89,90]. Levels of RFOs reported in faba bean were in the ranges 8–15, 1.1–3.9, and 4.4–13.7 g/kg, for verbascose, raffinose, and stachyose, respectively [91]. The content of RFOs in original flours and air-classified fractions were reported for red lentils, yellow lentils, green peas, and kabuli chickpeas [88]. Verbascose was only detected in green pea flour (36.74 mg/g, db.) and was lower in the coarse fraction (28.26 mg/g, db.) and higher in the fine fraction (59.50 mg/g, db.). Stachyose content was highest in red and yellow lentil flour, being 56.51 and 51.02 mg/g (db.), increasing to 73.3 and 87.5 mg/g (db.) in the fine fraction, respectively. On the other hand, raffinose content was highest in kabuli chickpea flour (39.9 mg/g, db.), increasing to 46.5 mg/g (db.) in the fine fraction. Raffinose content was lowest for green pea flour, and the respective coarse and fine fractions were 8.18, 7.76, and 10.34 mg/g (db.). Due to the prebiotic nature of these oligosaccharides, they have been reported to increase the bifidobacterial population in the gut microbiome, promoting the absorption of minerals, and enhancing the immune system while protecting against colon cancer [88].

Tannins, which are classified into hydrolysable tannins and condensed tannins, promote protein cross-linking. High-molecular-weight proanthocyanidins can precipitate proteins easily, whereas ellagitannins with rigid conformation tend to precipitate proteins less effectively [92]. Tannins are non-absorbable compounds that can generate health benefits in the gastrointestinal tract through radical scavenging and antiviral, antimutagenic, and antinutritional effects, during their colonic fermentation [93]. The legumes with the highest polyphenolic content are varieties with darker seed coats, such as red kidney beans and black gram (*Vigna mungo*). Condensed tannins (proanthocyanidins) have been found in the hulls of several varieties of faba beans and are also in coloured-flowered pea cultivars. Meanwhile, tannin-free and sweet seeds have been selected among broad beans, lentils, and lupins [94]. Tannins have been found at levels between 4 and 6 mg/g (db.) in faba bean cultivars [95]. Processing methods such as dehulling, soaking, cooking, autoclaving, fermentation, and extrusion have been effective in eliminating or reducing tannin content [96].

Other antinutrients include trypsin inhibitors, for which it has been reported that during digestion they inhibit the activity of enzymes such as trypsin, chymotrypsin, and pancreatic enzymes and reduce the digestion and absorption of protein. Trypsin inhibitor content ranges from 1.2 to 23.1 units/mg in different varieties of faba bean [91]. Saponins are mainly found in chickpeas, lentils, and beans, where a glycoside composed of a lipid-soluble aglycone consisting of a sterol (a triterpenoid structure) is attached to water-soluble sugar residues, which differ in their type and number [94]. Saponin content varies among cultivars, and under stress conditions, there is an increase in saponin content, which is related to transcriptional activation of the genes responsible for saponin biosynthesis through a signalling cascade involving salicylate and jasmonate [97]. Total saponin content in the seed coat of mung beans and adzuki beans was reported to be higher than in the cotyledons [98].

## 4. Pulse Protein Extraction

### 4.1. Dehulling, Splitting, and Milling of Pulses

Dehulling is the process of removing the hulls (seed coat), which predominantly contain insoluble dietary fibre (70–87%), comprising non-starch polysaccharides, such as cellulose, hemicellulose, and pectin [99]. Removal of the fibrous hull improves cooking time, flour quality, colour, digestibility, and palatability, leads to the removal of some antinutrients such as tannins, and increases the concentration of other antinutrients present within the cotyledon, such as phytic acid, and enzyme inhibitors, including those of trypsin, chymotrypsin, and α-amylase [99,100]. The dehulling process is efficient for pulses such as peas, lentils, and desi-type chickpeas but less efficient for kabuli-type chickpeas, faba beans, and mung beans, and their effect on dehulling efficiency has been reviewed [101,102,103,104].

### 4.2. Dry Fractionation

Dry fractionation, involving milling and air classification, offers a scalable, economical, and sustainable process for producing concentrated protein ingredients. This technique has since been applied to a wider range of pulses, with protein concentrations typically ranging between 40 and 65 g/100 g (db.) [105,106,107]. Pulses have organised cellular structures, with smaller protein bodies (1–3 µm) embedded within a matrix of larger starch granules (20–40 µm). Milling serves to disrupt these structures, disentangling the protein bodies (<10 µm) from the starch granules (20–40 µm) [107]. Factors such as milling intensity, seed moisture, and chemical composition can influence the degree of disentanglement [105].

Impact classifier milling produces flour of a fine particle size (top size of <45 µm) with low starch damage, which is suitable for the purpose of air classification. Impact classifier milling of faba bean, yellow pea, and red lentils has been shown to result in fine flours with minimal starch damage (1.3–2.4 g/100 g) [23]. Subsequent air classification separates protein (fine fraction) from starch (coarse fraction) respective protein concentrations ranged from 55.5 to 61.4 g/100 g (db.), with protein yields ranging from 34.0% to 47.2% (Figure 3).

### 4.3. Tribo-Electric Separation

Tribo-electric separation, used for flour and gluten–starch mixtures [108], charges particles through collisions in a gas flow, allowing separation via an external electric field. While effective for protein–starch separation, it generally results in lower protein content compared to air classification [3]. For example, tribo-electric separation of navy bean flour achieved 46.5% protein, whereas air classification reached 51% [108]. A three-stage process applied to lupin flour was shown to increase protein content to 65% but reduce yield by 6% [80]. In contrast, air classification of defatted lupin flour resulted in protein concentrates of 57% protein, but with a much higher yield (21%) [109].

### 4.4. Wet Fractionation

Wet fractionation methods involve the use of water or chemicals to solubilise proteins and separate the other seed constituents, such as starch and fibre. Wet fractionation methods are more energy-intensive and require a drying step to recover protein as a free-flowing powder [110]. The protein content of wet fractionated protein ingredients can vary from protein concentrates (60–70%) to protein isolates of >80% [111,112,113]. Figure 4 summarises commonly used methods to extract protein isolates from pulses. The method of protein extraction and different drying technologies can have implications on the structure and functionality of the protein, as it may cause denaturation or modification [112,114]. The advantages and disadvantages of dry and wet fractionation methods are described in Table 3.

### 4.5. Isoelectric Point

Isoelectric precipitation (IEP) is a common method for precipitating proteins, exploiting differences in protein solubility at various pH levels. Protein solubility is lowest at or near the isoelectric point (pH 4–5) and higher at more acidic or alkaline pH levels. The alkaline solubilisation and IEP method is commonly used for pulse proteins, requiring alkaline solutions usually prepared using NaOH (pH 8–11) [115]. Key factors such as temperature, time, pH, and substrate/solvent ratios must be optimised to maximise protein purity and yield [116]. The particle size of pulse flours (e.g., yellow pea, green lentil, kabuli chickpea, and navy bean) also affects protein concentration and yield [117]. After solubilisation, proteins are precipitated with acids such as HCl (pH 4–5) and separated by centrifugation where the protein is often washed and neutralised [118]. Alkaline solubilisation and IEP were used to prepare protein isolates from various pulses. For example, protein contents from mung bean (whole seed, raw dahl, roasted dahl, germinated) ranged from 87.3% to 90.4%, with protein concentrates ranging between 27.1% and 80.8% [40]. Similarly, protein isolates from lentil, mung bean, and yellow peas had protein contents ranging from 84.39% to 93.92%, with protein yields from 53.74% to 68.21% [119]. Lower protein yields are often linked to protein denaturation and aggregation during extraction or post-extraction treatments.

### 4.6. Ultrafiltration

Ultrafiltration (UF) is another commonly used technique for protein extraction and involves the use of membranes that have pore sizes or molecular-weight cut-offs that are smaller than the hydrodynamic size of proteins. Proteins are retained (retentate) whilst other lower-molecular-weight components permeate through the membrane and are separated [114]. Protein isolates were prepared from Sweet Lupin using alkaline solubilisation, followed separately by IEP and UF for comparison [120]. Alkaline solubilisation resulted in the solubilisation of 87% of the protein, in which 59% was recovered by IEP. On the other hand, ultrafiltration appeared more effective than IEP, recovering 92% of the total protein. The IEP lupin protein concentrate contained 671 g/kg protein in comparison to the UF protein concentrate, which contained 751 g/kg protein. Losses of the acid-soluble protein fraction during IEP resulted in a lower protein concentration and lower proportion of sulfur-containing AAs.

### 4.7. Micellisation (Salt Extraction)

The micellisation or salt extraction method involves two main steps: Firstly, the salt-soluble proteins (globulins) are solubilised in salt solutions such as sodium, calcium, and potassium chloride at neutral pH 7 (10–60 min) at moderate temperatures (15–35 °C) and concentrations of 0.2–0.8 M. The soluble proteins in the salt extract are subsequently recovered by reducing the ionic strength via membrane separation (dialysis) or precipitation by dilution with water at a lower temperature (1–4 °C) [121].

### 4.8. Modification and Improvement of Pulse Proteins

Fermentation has the potential to improve the quality, functionality, and value-adding of pulse protein ingredients. Enzymes produced by microorganisms during fermentation break down proteins into amino acids and peptides and other water-soluble products of protein denaturation. Air-classified pea protein concentrates (52%) subjected to fermentation for 120 h using six different fungal organisms (*Aspergillus niger*, *Aspergillus oryzae*, *Aureobasidium pullulans*, *Neurospora crassa*, *Rhizopus microspores* var. *oligosporus*, *Trichoderma reesei*) have been shown to increase total phenolic content, protein content, protein solubility, saponin content, and fibre fractions [122]. The authors also reported that fermentation improved the flavour profile of pea protein concentrates. Hydrolytic enzymes of this strain cause the breakdown of the cell wall matrix and subsequently improve the saponin extractability.

Protein isolates have been obtained from germinated seed material of yellow pea (Celine) using alkaline solubilisation (pH 9) and IEP (pH 4.5) [123]. Seed samples were analysed after the soaking stage at days 0, 3, and 6. Seeds were oven-dried, then dehulled and milled to flour. Results showed that protein isolates increased after the germination process from 83.6% to 87.9% protein, but protein yields decreased from 18.6% to 14.3%. Germination of pulses has been shown to reduce levels of stachyose, raffinose, and trypsin inhibitors [124] and increase levels of protein and vitamins [125]. Increases in the in vitro protein digestibility and bioavailability of minerals have also been reported [122]. Alkaline solubilisation and IEP have been used to obtain protein isolates of 88.4%, and protein yield of 65.9%, from germinated seed material from a mung bean variety [40].

## 5. Functional Properties of Pulse Proteins

Functional properties of proteins are defined as the properties that can affect the behaviour of the proteins in food systems during processing, preparation, storage, and consumption. These can be classified into three groups based on the mechanism of action: (i) properties related to hydration (solubility, and water- and oil-holding capacities), (ii) properties related to the structure and rheological characteristics (gelation), and (iii) properties related to the protein surface characteristics (emulsification and foaming). Table 4 [99,126,127,128,129,130,131,132,133,134,135,136,137] summaries the functional properties of pulse proteins. The factors influencing the functional behaviour of proteins are related to their size, structure, amino acid composition and sequence, hydrophobic/hydrophilic structure and ratio, and molecular rigidity in response to environmental factors [138].

### 5.1. Solubility

Solubility is one of the most crucial properties of protein ingredients and can have a significant impact on many other functional properties. Many food applications require high solubility, particularly for beverages, where sedimentation of insoluble protein is undesirable [139]. The overall solubility of proteins depends on the balance of protein–protein and protein–water interactions as salts in the solution may shield electric charges and diffuse surrounding proteins; they may also affect the solubility by influencing the surface charge and electrostatic repulsive forces, depending on the concentration and type of salts [140].

Plant proteins can be characterised or classified based on their ability to solubilise in different solutions. The major fractions of proteins in pulses are globulins (50–70%), followed by albumins (18–25%) [141]. Albumins are soluble in water, globulins in salt solution, prolamins in aqueous alcohol, and glutelins in acid or alkaline solution [142]. Therefore, protein solubility is impacted by external factors including ionic strength, pH, and combination of solvents but also by the protein’s state of denaturation. In its native state, a protein exposes its hydrophilic groups on the surface, which increases its solubility. With increased denaturation and aggregation, the protein unfolds and becomes less soluble. These structural changes often occur during the wet fractionation of commercial protein isolates, which could make technologies such as air classification more attractive for producing protein concentrates that retain their native state [132].

The solubility of air-classified faba bean protein concentrate has been compared to that of wet-fractionated faba bean protein isolate [126]. It was found that air-classified faba bean protein concentrate showed higher protein solubility at pH 4.5 (15%) and pH 7 (87%) when compared to the wet-fractionated faba bean protein isolate at pH 4.5 (3%) and pH 7 (31%). The improved solubility of air-classified protein concentrates could be exploited in different food applications including plant-based meat and dairy alternatives, as well as a range of beverages, including sports drinks.

### 5.2. Water- and Oil-Holding Capacity

Water- and oil-holding capacities refer to the amount of water or oil that can be absorbed per gram of protein and determine the texture and mouthfeel of foods. Water-holding capacity (WHC) is an important functional property in soups, doughs, and baked foods, which are supposed to absorb water without dissolution of the proteins, thereby providing thickening and viscosity properties [131], whilst oil-holding capacity (OHC) is required in ground meals, meat analogues, and extenders, in which oil contributes to the texture and mouthfeel of food products [143]. In contrast, OHC usually increases with denaturation, as the proteins’ hydrophilic core is exposed, thus favouring oil-binding [144].

Both WHC and OHC influence the textural and sensory properties of foods in various ways. The WHC must be considered when designing plant-based hydrocolloid food products to ensure they retain moisture during production and shelf-life. Plant-based yoghurts must be able to retain water over time to avoid phase separation, which is considered unappealing by consumers [145]. WHCs in mung bean, yellow pea, and cowpea protein concentrates produced from dry fractionation have been determined [132]. This study reported that the WHC for mung bean, yellow pea, and cowpea were 2.1, 1.53, and 2.06 g/g, respectively. Faba bean and pea protein isolates prepared using alkaline solubilisation and isoelectric precipitation (1.8 and 1.7 g/g, respectively) gave similar results [10], which suggests that WHC and OHC of air-classified protein concentrates are comparable to protein isolates produced from wet fractionation.

### 5.3. Emulsifying Properties

An emulsion is a suspension of two immiscible liquids (mainly oil and water), in which one is dispersed in the form of droplets within the continuous phase of another liquid [53]. The interfacial surface tension between these two liquids causes a thermodynamically unstable system inside the emulsion requiring stabilisation by emulsifiers. This stabilisation delays or stops structural changes, including coalescence, creaming, flocculation, and sedimentation. This effect is commonly measured over emulsifying ability and capacity. Pulse proteins can act as plant-based emulsifiers due to their surface activity, which allows these proteins to stabilise emulsions in plant-based foods, such as dairy analogues and dressings [145].

Physicochemical and emulsifying properties of pea, chickpea, and lentil protein isolates have been investigated [146]. The authors reported that the emulsifying properties of the isolates (droplet size distribution, flocculation, coalescence, and creaming) were correlated as a function of pH values. The three protein isolates showed similar physicochemical properties, such as good solubility and high thermal stability despite a high degree of denaturation. The influence of pH on the stability of oil-in-water (O/W; 10/90 wt%) emulsions stabilised by the protein isolates showed that a significant improvement in emulsion stability was observed as the pH values depart from the isoelectric point (pI = 4.5). The emulsifying properties of pea, chickpea, and lentil protein isolates, such as droplet size distribution, flocculation, coalescence, and creaming, have been reported [147].

### 5.4. Foaming Properties

Foods in the form of edible foams have existed in human diets for thousands of years [148] and are becoming increasingly popular among consumers due to their visual appeal and sensory properties [149]. They are available in a variety of formats, including liquid (beverages), semi-solid (ice cream, whipped cream, aerated desserts), and solid (bread, cakes, breakfast cereals, and aerated chocolate bars) [150,151]. Foams consist of a gas phase that is dispersed in an aqueous or solid phase. When aerated, the surface-active protein molecules in the dispersion form a film around the gas and thus create the foam structure. When a foam is formed, proteins unfold and re-orient due to their surface activity. Their hydrophilic groups orientate towards the aqueous phase, while their hydrophobic groups orientate towards the gas phase. This protects the gas bubbles and stabilises the foam structure [152].

Foaming properties are commonly measured in terms of foam capacity and foam stability. Different methods are used to measure these properties, but generally, protein solutions are homogenised at high speed to produce the foam, and the foam capacity is expressed as the volume increase (%) due to whipping. The foam stability is the change in the foam volume over a set period (e.g., 60 min) [153].

The foaming capacity for yellow pea (Delta), green pea (CDC Striker), lentils (large green lentil, CDC Plato, small green lentil, CDC Viceroy, and red lentil, CDC Blaze), navy beans (Envoy), and chickpeas (kabuli chickpea, CDC Frontier, kabuli chickpea, B90, and desi chickpea, CDC Myles) has been determined [154]. The navy bean protein concentrate had the highest foam capacity (622%), and chickpea had the lowest (26%). Lower foam capacity for chickpea protein concentrates is mainly due to higher fat content compared to other types of pulses, which in turn limits the foam capacity.

The foam capacity of protein isolates prepared from different commercial Indian chickpea varieties was found to range between 30.4% and 44.3% and was higher (*p* < 0.05) for the protein isolates compared to their original flours [155]. Foam capacity also increased when the amount of protein isolate increased in the aqueous dispersion, allowing for a more stable foam formation. There was also a rapid increase in foam volume up to 7% (*w*/*v*) solids concentration with a maximum of 10% (*w*/*v*) when the amount of protein isolate was increased. Foam stability showed that among the pulses, the kabuli chickpea protein isolate had the highest foam stability (94.7%) after 120 min of storage, indicating that proteins that are soluble in the continuous phase (water) are more surface-active.

### 5.5. Gelation

Gelation is an important functional property for many different food products, including soups, puddings, jellies, and meats. Gels are formed when protein solutions transform into semi-solid structures via unfolding, association, and aggregation of proteins [153]. These three-dimensional, cross-linked protein networks are structured isotopically (direction-independent). There are various techniques to induce gelation including heat, pressure, pH shifting, salt addition, or enzymes, each of which has different underlying mechanisms [156,157]. Gelation temperatures of pulse proteins are generally dependent on their thermal stability and are higher than the denaturation temperature of proteins, as denaturation is the prerequisite for heat-induced gelation. The least gelling concentration (LGC) is used as an important index of gelling capacity, and it can be defined as the lowest concentration needed to form a self-supporting gel. Gelation capacity can be improved by the presence of a moderate NaCl concentration and at low ionic strength [158].

Heat-induced gels from protein isolates prepared from lentils, mung beans, and yellow peas have been produced and characterised [129]. Among these, mung bean protein isolates showed lower gelation temperature (T_gel_ 77 °C) and developed stronger gels compared to lentil and yellow pea protein isolates. The effect of salts (0.5 M NaCl or 0.25 M CaCl_2_) and protein concentration (7.5–15%) on the gel-forming abilities of lentil, yellow pea, and faba bean protein concentrates formed at pH 7.0 has been determined [25]. The surface hydrophobicity of yellow pea (84.8 arbitrary units, a.u.) was found to be lower than lentil (147.2 a.u.) and faba bean (135.0 a.u.). On the other hand, the surface charge for lentil, yellow pea, and faba bean concentrates was −37.8, −28.4, and −29.3 mV, respectively. The Lg/Vn ratio of yellow pea was 0.65, lentil (0.57), and faba bean (0.41). The presence of salts reduced the least gelling concentration. Lentils and faba bean protein concentrates were shown to have a more organised structure than yellow peas when they were analysed by Confocal Laser Scanning Microscopy (CLSM). The network appeared more ordered as the protein concentration increased or in the presence of NaCl or CaCl_2_ according to CLSM and synchrotron-based micro-computed tomography (µCT).

## 6. Uses of Pulse Proteins in Foods

Pulses have played a crucial role in the global diet for millennia and are a staple source of protein in many regions. However, it has been reported that less than 4% of people aged two years and older meet the recommended intake of vegetables and legumes (pulses) per day (ABS, 2016) [159]. A survey (*n* = 505) was conducted on consumers’ perceptions to investigate the consumption, knowledge, attitudes, and culinary use of pulses in Australia [160]. For consumers (47%; 177/376), chickpeas, green peas, and kidney beans were most often consumed and in general, pulses were consumed 2–4 times weekly. Consumers also identified pulses as a source of protein and dietary fibre as key nutritional qualities. For non-consumers of pulses (7%; 34/463), factors such as taste, limited knowledge on how to prepare and include pulses in meals, and the time taken to prepare them, as well as family preferences, hindered consumption. Food manufacturers need to focus on developing and reformulating pulsed-based food products that are convenient, tasty, affordable, and aligned with Dietary Guidelines. The Grains and Legumes Nutrition Council (GLNC) actively promotes the use of pulses to consumers and food manufacturers through evidence-based information [161]. The GLNC recently reported that the sale of plant-based meat and dairy substitutes has increased by nearly 30% from 2018–2019 to 2022–2023, with the plant-based milk category being the most developed and driven by innovation (GLNC; State of the Industry: An overview of plant-based products 2022). Table 5 summarises the different uses of pulse proteins in the food industry.

### 6.1. Bakery Products and Pasta

Celiac disease is an autoimmune disorder triggered by gluten intake, restricting people affected by this condition from consuming foods made from cereals such as wheat, rye, and barley. Gluten-free substitute foods can mitigate this, but they pose technological challenges to create products with the functional and sensory attributes of their traditional counterparts and can also increase the cost of food formulations [171]. Pulse proteins have the capacity to act as functional ingredients as they can be used as foaming agents, emulsifiers, water and oil binders, or gelling agents, which are all important properties in the production of bakery products, noodles, and pasta [172].

In many bakery applications, it is essential that the dough has enough elasticity and extensibility to allow gas retention during expansion, which creates the typical volume and cell structure of breads and other baked goods [153]. Wheat gluten has excellent capability to create fibrous networks with such properties, due to its molecular structure [141]. However, there are various opportunities for the inclusion of pulse protein ingredients into bakery products. Because pulse proteins do not develop a gluten network, it is difficult to completely substitute wheat flour with pulse flour. On the other hand, pulse proteins can be combined with starches or with other different sources of pulse proteins. The presence and mixing of other ingredients, in particular sugar and aromatic ingredients, but also fat or oil, helps to mask the off flavours of pulse concentrates, making them more acceptable in bakery products [70]. A study showed that when pea protein isolate (87% protein) was added at two inclusion levels (1% and 6%) in wheat dough, there were improvements in the rheological, structural, and nutritional properties [173]. The addition of protein isolate in the dough decreased the surface tension of the colloidal system, and the foam bubble size was reduced during bread mixing, which led to improved distribution of gas cells and improvement in the overall texture.

The effect of blending chickpea, broad bean, common bean, and red lentil flours at different inclusion levels of 5%, 10%, and 15%, with wheat–rye flour for bread making has been determined [174]. Higher enrichment levels of protein decreased bread volume, increased resistance to starch retrogradation, and improved aroma and flavour. The inclusion of 30% of chickpea protein concentrate with wheat flour in sourdough breadmaking was found to increase protein content and decrease specific bread volume, leading to the formation of a denser crumb structure [175]. The effects on bread quality (lower bread volume) were related to the inferior formation of a gluten network in the bread. The addition of faba bean protein isolates (90%) or concentrate (60%) in cracker dough was shown to improve the overall nutritional content [162]. Higher protein, dietary fibre, and resistant starch content were found in crackers with the addition of faba bean protein, compared to the wheat-based cracker formulations.

The effect on quality parameters in pasta using different ratios of pea and faba bean protein isolates, which were 100:0, 75:25, 50:50, 25:75, and 0:100, has been studied [176]. It was observed that both protein isolates provided extrudability by forming a uniform product at rates above 400 g/min and reaching up to 527.1 ± 2.5 g/min in the 100% faba bean protein isolate formulation. An overall change in colour was observed, and hardness increased especially at high rates with pea, water uptake, and cooking losses. The effect of adding pea and soy protein isolates in the formulations of pasta has been determined [177]. Results showed improvements in chemical score, digestible indispensable amino acid scores, colour, and firmness.

Lentil protein concentrates have been reported to be used as egg protein and milk replacers in donuts [115] The addition of lentil protein concentrates in the donut dough improved cooking characteristics, which reduced the loss of moisture and decreased the rate of hardening of donuts during storage but reduced the overall sensory score. Similar results were reported when lentil protein concentrates were used to replace egg proteins or milk in angel food cake [116]. The overall sensory acceptance was reduced, but for muffins, the complete substitution of proteins for lentil protein concentrates improved the sensory score when compared to traditional angel food cake. The inclusion of pea protein isolates at substitution rates of 0–30% in cookies increased water absorption capacity and dough rheological parameters (G’ and G” values) but did not improve the spread ratio or hardness [178]. However, blending with 10% of lupin protein concentrate improved the surface area and hardness of cookies [179].

The three main drivers behind the development of pulse-based bakery products have been described as improving the nutritional value of proteins (protein quality), creating gluten-free products, and replacing animal proteins in formulations [171]. Achieving a balanced amino acid composition for proteins in food is essential to the human diet. Pulses are high in lysine and low in sulfur-containing amino acids, namely, methionine and cysteine, whilst cereals such as wheat are low in lysine and high in sulfur-containing amino acids; thus, blending of these complementary sources of protein improves protein quality in diets [180].

The physicochemical and sensory properties of vegetarian pasta produced with semolina flour and the addition of pea protein isolate at different rates of 0, 10%, 20%, and 30% have been determined [181]. The stated author reported that sensory analysis revealed that the addition of 10% pea protein isolate was the most preferred and the satiety lasted longer when consumers consumed pasta containing 10% pea protein powder compared to the control pasta. Physicochemical properties (volume, cooking time, colour parameters, and weight loss) were improved with the addition of 10% of pea protein isolate when compared to other rates of pea protein isolate addition and the control.

### 6.2. Dairy Alternatives

Processed dairy products from cows’ milk, such as cheese, milk-based drinks, and yoghurt, as well as isolated whey protein, offer an accessible source of high-quality protein with a complete amino acid profile and unique sensory properties [182]. There is significant interest in replacing animal-based products with plant-based dairy alternatives [172], which is reflected by the increasing number of products available in supermarkets. However, it is crucial that these alternative products offer the same nutritional quality equivalent to cow’s milk, and also that they closely mimic sensory attributes such as visual appearance, mouthfeel, and flavour [183].

There is significant potential for using pulse proteins in the replacement of cow’s milk, as pulse proteins provide a range of health benefits, are more sustainable, and alleviate ethical concerns of a growing number of consumers in relation to animal welfare [184]. When other plant-based milk alternatives, such as those made from cereals and nuts (almonds), are compared to cow’s milk, lower values of protein content are found (0.1–1.5%) compared to cow’s milk (3.3–3.5%). Pulse-based milk obtained from sweet lupin and chickpea protein concentrates have been reported [167]. The protein content of plant-based milk made from chickpea and lupin was 3.24% and 4.05%, respectively, having similar protein content to cow’s milk. Another study reported that lupin and chickpeas can be used as pulse-based milk alternatives [185]. Lupin protein contents in these products ranged from 1.8% to 2.4% and between 1.0 and 1.5% for chickpeas. Sensory analysis showed that chickpea milk resulted in a better taste profile when compared to lupin.

Plant-based yoghurt alternatives, such as those made from soy and coconut, have been commercially available to consumers over the last few years, whilst pulse-based yoghurts are more recent but with a very limited number of products and availability [186]. Pulse-based yoghurts can be manufactured from water extracts of whole pulses or from a blend of pulse protein ingredients, fat, water, and sugar, followed by fermentation [186]. Pea protein isolate was used to produce yoghurt, which involved heating steps at 80 °C (30 min) and 60 °C (1 h), with a protein concentration of 10 g/L being enough to produce gels in subsequent acidification via lactic acid fermentation [187]. For lupin-based yoghurt alternatives, ultra-high temperature treatment at 140 °C (60 s), with a 2% lupin protein isolate solution prior to fermentation, produced a yoghurt with higher viscosity and more firmness than a common pasteurisation process performed at 80 °C for 60 s [168].

Pulse-based cheese alternatives, the production of which involved blending of zein protein isolate (maize) and chickpea protein concentrates at different ratios of 0:100, 25:75, 50:50, 75:25, and 100:0, respectively, have been described [166]. Cheese quality parameters such as stretchability, meltability, texture properties, rheological properties, thermal behaviour, and microstructure, were improved when zien and chickpea were blended at a ratio of 75:25, respectively.

The techno-functional properties of protein isolates of broad beans, mung beans, and lentils in dairy products have been reported [188]. Broad bean and mung bean protein isolates showed higher functionality properties like emulsification, foaming, and oil and water absorption properties when compared to lentil protein isolates in dairy products.

### 6.3. Meat Analogues

With the growing consumer interest in reducing meat intake, plant-based meat analogues have become increasingly popular. Meat analogues aim to closely mimic the texture and flavour of animal meat products, to create a similar sensory experience for the consumer [189]. A key component of meat analogues is to form anisotropic, fibrillar structures that resemble animal muscle tissue. There are various techniques to create such structures, which resemble fibres, but extrusion-based methods are easily scalable for commercial production [190]. Other texturisation methods include shear cell, freeze casting, spinning, and self-assembly, but their application is currently limited to lab-scale or initial scale-up [191,192]

Extrusion can be divided into low- and high-moisture extrusion processes, both of which are used to produce texturised vegetable proteins (TVPs). Low-moisture extrusion processes protein doughs with less than 50% water content, whereas high-moisture extrusion can process dough with more than 50% water content. Low-moisture extrusion is a well-established technique for producing meat analogues [190], generating TVP, which differs from high-moisture extrudates by its expanded structure. The process involves a sudden pressure drop as the material exits the extruder die, triggering water evaporation, cooling, increased viscosity, and transformation of the material to a glassy state. The evaporation creates a platform for bubble expansion and stabilises the material structure. TVP is commonly used as a raw ingredient for the production of meat analogues but requires rehydration and further processing to form the basis for various plant-based products, such as nuggets, chucks, and strips [193].

High-moisture extrusion is a more recent technique, which is used to create meat analogues with meat-like fibrous structures and desirable sensory properties. This process involves combining proteins with other components at low temperatures, followed by heating and shear forces that result in denaturation and aggregation of proteins. This process leads to the formation of fibril-like structures through the formation of hydrogen and disulfide bonds. The cooling die in the extruder’s final section aids in shear alignment and stabilisation, promoting fibre formation. This technique offers versatility in producing various fibrillar textures, including whole-cut resembling extrudates [194].

Shear cell technology was developed from a rheometer design and employs a cone-in-cone cell. The cell exerts uniform shearing forces on a pre-mixed blend of plant proteins, salts, and additives. This method creates nano- and micro-scale fibrous structures, which resemble meat and fish cuts in size and texture. In 2022, there was only one start-up company which utilises this technique for limited commercial production of meat analogues [193]. Freeze-casting achieves fibrous formations by dispersing proteins within an aqueous solution and freeze-concentrating them through directional heat removal. Ice crystal expansion reconfigures the proteins from isotropic to anisotropic alignment. By evacuating water, the structure then solidifies the oriented fibres via heat treatment. This process is applicable to water-insoluble plant protein fractions. The blending of these proteins with water generates a mouldable dough, mechanically stretched to develop anisotropic structures. The widely embraced seitan, a wheat gluten-based meat analogue that is popular in various Asian cultures, is based on this approach [157]. Two different spinning approaches yield fibrous protein architectures on a laboratory scale. Electrospinning propels a protein solution from a high-voltage needle or spinneret, forming a nano-fibre upon solvent evaporation.

The production of palatable meat analogues using high-moisture extrusion cooking is a complex process that depends on the source of protein and the extrusion conditions used. The effect of high-moisture extrusion on the raw material characteristics, extruder responses, and texture properties, using three commercial pea protein isolates, has been reported [170]. Pea protein isolates showed similar chemical compositions, but their functional properties affected the viscosity of the protein mass during the initial heating phase of the extrusion process. The product texture properties depended on the cooking temperature and were similar among the proteins, although different energy input was observed during texturisation. Furthermore, pea protein isolates developed a fibrous structure similar to fibrous whole muscle, with potential for producing a range of products to meet different consumer requirements. Similar results, in which blends of faba bean protein concentrate and isolate were used as raw materials for high moisture extruded meat analogues, have been found [169].

The physicochemical and sensory qualities of pulse-based meat with dry fractionated pea protein concentrate and with pea, oat, and soy protein isolates have been explored [88]. A low-moisture extrusion process was subsequently applied and samples containing pea protein concentrate showed lower protein content, high fat content, and high carbohydrate content, shorter rehydration time, and lower cohesiveness, hardness, springiness, and elastic behaviour, when compared to oat and soy protein isolates. On the other hand, sensory characteristics for samples containing pea protein concentrate had more intense taste and odour profiles when compared to protein isolates having more neutral profiles.

The use of wet and dry fractionated pea protein ingredients for production of meat analogues has been investigated [195]. Higher resistance to elastic and viscous deformation, darker colour, and lower textural properties (hardness and chewiness) were observed for the dry-fractionated pea protein-based analogue, when compared to wet-fractionated pea protein-based analogue, which had higher protein content, greater hardness, and chewiness. These differences are related to the processing method, as the dry-fractionated pea protein had more starch, phenolic compounds, and pigments, which contributed to the softer texture and darker colour when compared to wet-fractionation ingredient.

Textural and sensory properties of deep-fried meatball analogues produced with different rates of cowpea, yellow peas, green gram, and horse gram protein isolates have been reported [196]. Meatballs were evaluated at 20:20:20:20, 20:15:15:10, and 10:15:10:15 (green gram protein concentrate–horse-gram protein concentrate–cowpea protein concentrate–yellow pea protein concentrate, respectively). Protein content of pulse protein isolates ranged from 79% to 87%. Colour and texture parameters (fracturability, hardness, cohesiveness, and adhesiveness, springiness, gumminess, resilience, and chewiness) were evaluated and revealed that meatballs formulation 20:20:20:20 (green gram protein concentrate–horse-gram protein concentrate–cowpea protein concentrate–yellow pea protein concentrate, respectively) showed similar properties to control samples (chicken) when compared to other two formulations suggesting that when different composite pulse proteins are used, it is possible to mimic conventional chicken meatballs’ colour and texture properties.

The use of pulse proteins in food formulations presents many different challenges. Variability in protein content and nutritional quality across different pulse crops makes consistent product outcomes challenging. Antinutritional factors and potential allergens of pulse proteins can affect food safety and protein digestibility. Additionally, the off-flavours associated with pulse-based products require further improvement, either through plant breeding initiatives, use of pre-treatments, or masking additives [194]. Overcoming variations in colour and appearance, especially in comparison to traditional meat products, is another challenge. Addressing these will require innovative solutions, such as improved sourcing strategies, advanced processing methods, and the application of flavour masking and colour enhancement techniques [197].

## 7. Future Trends

Pulse proteins are a promising and sustainable alternative to animal proteins, providing a good source of nutrition, unique functional properties, and suitability for use in a wide range of foods. Considerable efforts are still needed to increase consumer awareness of the health benefits of including pulses in their diets, which will lead to greater demand for pulse-based foods. Dry- and wet-fractionated protein ingredients are commercially available for a range of pulses, but there is a great opportunity to use new and existing technologies to improve protein concentration and yield of these ingredients, thereby increasing the cost-effectiveness of commercial production. Further research on hybrid fractionation methods is required to better understand the benefits of this approach, in terms of sustainability, process efficiency, innovation, and value-adding of functional pulse protein ingredients.

Pulse protein functionality is crucial for new food product development; however, functional properties can vary between pulses and their many varieties, using different processing conditions, as well as seasonal variations. This scenario presents a challenge for the food industry and requires the development and adoption of standardised testing methods to ensure consistency in food product quality. The presence of off-flavours limits the use of pulses in foods and is another challenge that needs to be overcome. Currently, pea protein is the most widely used source of pulse protein commercially, and further research is required to leverage a wider range of pulses, including faba bean, mung bean, lentils, chickpea, and lupin. Future plant breeding initiatives could also be targeted towards developing new and improved varieties that address some of the limitations of pulses and help meet the needs of food manufacturers. The current status, limitations, and future development of pulse proteins for food applications are summarised in Figure 5.

Also of significant importance is the value-adding of starch and fibre co-products generated from the fractionation of pulse protein ingredients. Typically, these co-products are being used for traditional and lower-value animal feed uses, but there is significant potential for utilising these co-products, and other side-steams (e.g., steep water) from wet fractionation, for new and innovative applications. Functional and innovative starch and fibre ingredients can also be used in a range of food applications, or for pet foods, and even in a diverse range of pharmaceutical and industrial applications [198,199]. Some of the more conventional industrial applications include biofuels, textiles, and bioplastics, as well substrates for industrial fermentation and enzymatic processes [18].

## 8. Conclusions

Pulse proteins are set to play a key role in the future of food by offering sustainable and nutritious alternatives to animal-based proteins. There is significant opportunity for pulses to help meet this demand, through processing of value-added protein ingredients with desirable functional properties. Understanding the functional properties of pulse proteins, and how to exploit them, is vital for their utilisation in a wider range of food applications. With their high protein content and essential nutrients, pulses can enhance the nutritional quality of foods while supporting environmental sustainability. As demand for plant-based protein ingredients increases, pulse proteins will continue to be incorporated into a wide range of food products, which appeal to consumers, thereby contributing to healthier diets and greater food security. This in turn will drive investment in plant breeding initiatives to deliver new and improved pulse varieties, tailored towards the needs of food manufacturers. Future research supported by innovative and collaborative partnerships between the public and private sector will help overcome the challenges and limitations of pulses and unlock their full potential in the global food landscape.

## Figures and Tables

**Figure 1 foods-14-01151-f001:**
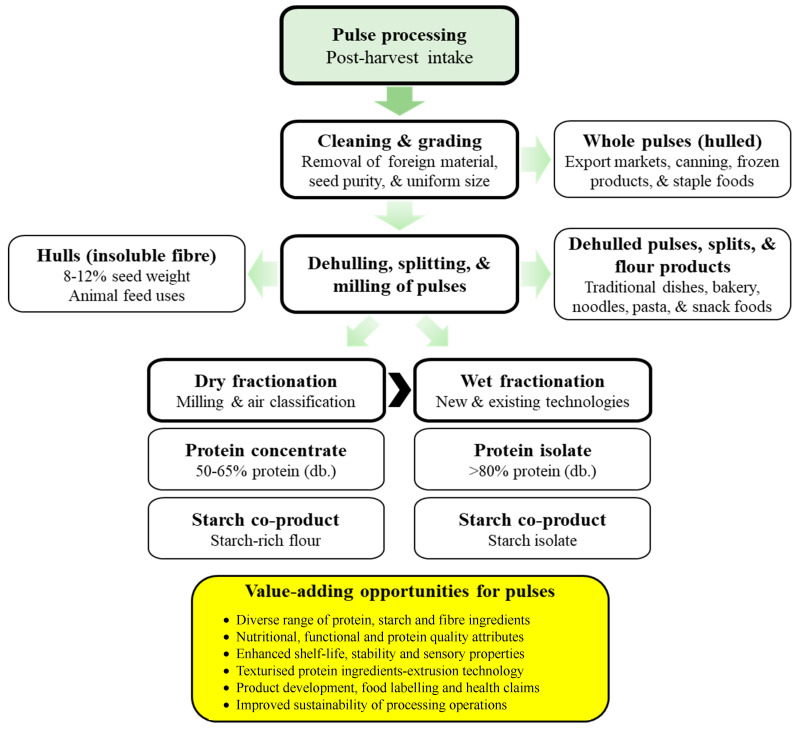
Pulse processing value chain and value-added opportunities for maximising the use of pulses and their ingredients.

**Figure 2 foods-14-01151-f002:**
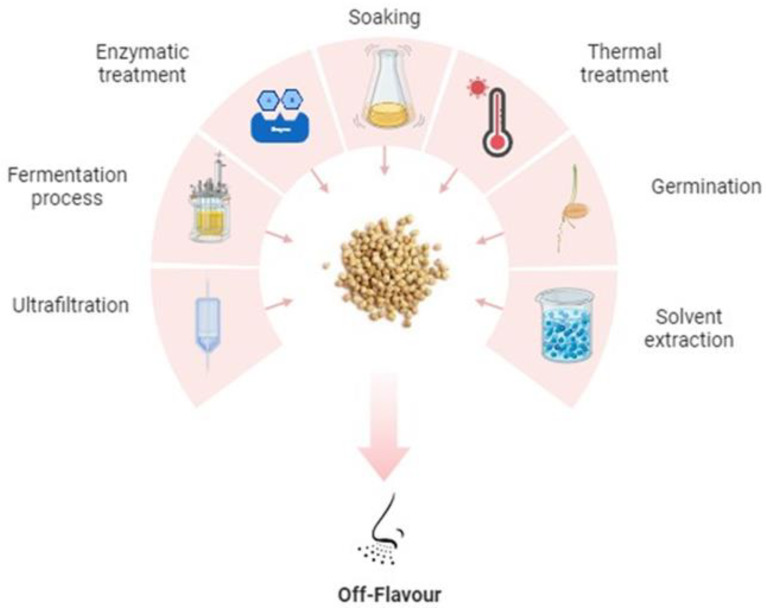
Processing techniques used to remove or modify off-flavours in pulses (Adapted from [70]).

**Figure 3 foods-14-01151-f003:**
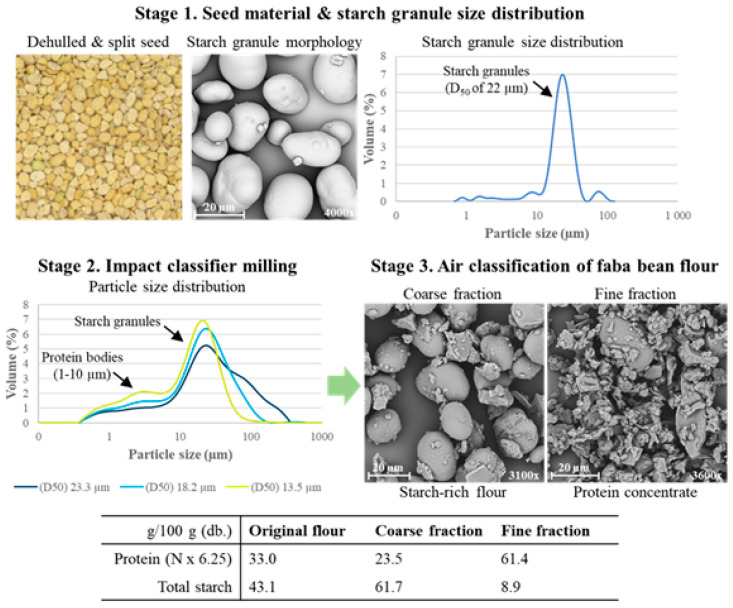
The main stages of the dry fractionation process are highlighted for the Australian faba bean, including (1) seed pretreatment (dehulling and splitting), (2) impact classifier milling, and (3) air classification for separation of coarse and fine fractions (Adapted from [36]).

**Figure 4 foods-14-01151-f004:**
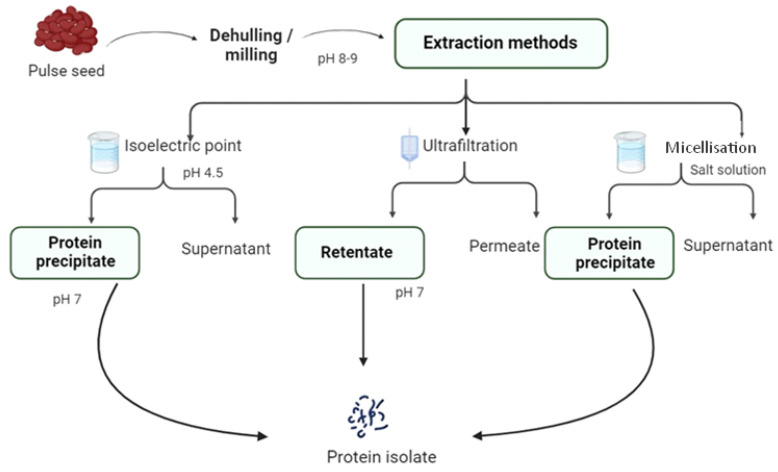
Schematic representation of isoelectric precipitation, ultrafiltration, and micellisation techniques commonly used to extract protein isolates from pulses (Adapted from Gunes & Karaca, 2022 [112]; Kornet et al., 2021 [114]).

**Figure 5 foods-14-01151-f005:**
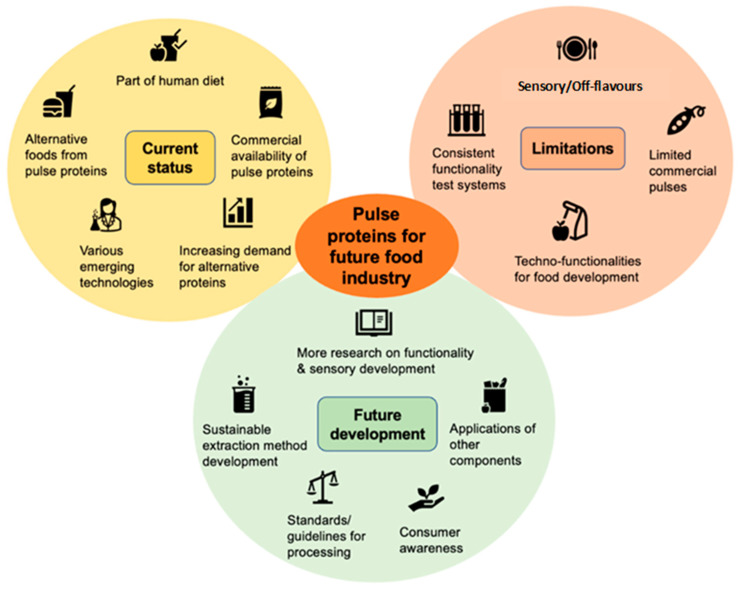
Current status, limitations, and future development of pulse proteins for food applications.

**Table 1 foods-14-01151-t001:** Annual worldwide pulse production over the four-year period from 2018 to 2022 [18].

Pulse Production (mmt)	2018	2019	2020	2021	2022
Beans, dry	27.5	25.6	27.4	27.7	28.5
Chickpeas, dry	16.9	14.2	15.1	15.9	16.5
Peas, dry	13.4	14.0	14.7	12.4	13.0
Cowpeas, dry	8.4	8.5	9.0	9.0	9.2
Broad beans and horse beans, dry	5.6	5.4	5.7	6.0	6.2
Lentils, dry	6.6	5.8	6.5	5.6	6.0
Pigeon peas, dry	5.4	4.4	5.0	5.5	5.8
Other pulses (*n.e.c.)	4.8	4.3	4.6	4.7	5.0
Lupins	1.2	1.3	1.0	1.4	1.5
Vetches	0.6	0.7	0.7	0.6	0.7
Bambara beans, dry	0.2	0.2	0.3	0.2	0.3

*n.e.c.: not elsewhere classified.

**Table 2 foods-14-01151-t002:** Proximate composition of pulses grown around the world.

Common Name	Latin Name	Country	Cultivar	Protein (%)	Fat (%)	Carbohydrate (%)	Minerals (%)	Fibre (%)	Reference
Chickpea	*Cicer arietinum*	India	Chana	18.5	4.3	55.0	3.5	4.0	[20]
		Canada	CDC leader	20.9	6.5	65.4	3.0	4.3	[21]
		Ethiopia	Natoli	17.7	5.2	62.4	3.7	6.7	[22]
		Australia	Desi	15.7	4.4	-	3.7	-	[23]
Kabuli chickpea	*Cicer arietinum*	India	Kabuli chana	20.5	5.7	55.0	3.8	4.0	[22]
		Italy	MG_1	17.4	3.7	54.3	3.3	5.5	[24]
		Canada	CDC Orion	20.0	6.6	43.2	2.9	-	[25]
		Australia	Kimberley Large	18.2	5.9	52.6	2.9	16.5	[26]
Lentil	*Lens culinaris*	India	Masoor	25.0	1.0	59.0	2.0	1.0	[23]
		Iran	Red	25.9	2.7	59.0	3.6	2.7	[27]
		Canada	Roxy	30.4	1.5	65.0	3.1	-	[28]
		Australia	Matilda	32.6	2.8	-	-	-	[29]
Field pea	*Pisum sativum*	India	Pant pea 25	20.3	2.0	58.5	2,9	3.4	[30]
		Pakistan	Paniola	20.0	1.7	89.7	-	0.8	[31]
		Ethiopia	Bilalo	23.9	2.7	59.2	3.4	3.6	[32]
		Australia	Commercial	20.7	0.3	4.7	0.5	25.1	[33]
Faba bean	*Vicia faba*	India	Bell bean	26.1	1.5	58.3	-	2.5	[23]
		Ireland	Victor	28.0	1.6	40.9	3.4	13.8	[34]
		Ecuador	Commercial	22.5	0.6	63.7	1.3	2.1	[35]
		Australia	Commercial	33.0	1.5	-	2.4	-	[36]
Mung bean	*Vigna radiata*	Nigeria	Wilczek	24.1	1.9	55.7	3.0	5.0	[37]
		China	Bailv 522	26.2	1.3	56.8	3.8	3.6	[38]
		India	AKM-8802	21.9	-	41.9	3.6	-	[39]
		Australia	Dahl	27.6	1.9	46.5	3.5	10.6	[40]
Lupin	*Lupinus albus*	Ethiopia	Commercial	34.9	1.3	50.5	3.3	9.9	[41]
		Ethiopia	Commercial	28.6	9.4	-	3.0	9.7	[42]
		Brazil	White	34.1	9.9	14.9	2.6	38.4	[43]
		Australia	Jenabillup	38.6	7.1	6.7	3.5	36.1	[44]

**Table 3 foods-14-01151-t003:** Advantages and disadvantages between dry and wet fractionation methods.

Method	Advantages	Disadvantages
Wet fractionation	High protein purity (>80%) Possible use of enzymes to enhance protein digestibility and functional properties Reduction in antinutritional factors—"washed out” Flavour modification—neutral to mild flavours	High water usage Energy consumption (drying) Chemical use (pH) Loss of native protein functionality Can be more expensive—equipment, electricity, storage, and waste management Microbial issues Scalability
Dry fractionation	Native protein functionality retained Sustainable—no water usage or drying stages, and less energy consumption No chemicals required No waste products Less equipment More economical Scalability	Moderate protein purity (50–65%) Concentration of some antinutritional factors No flavour modification—beany, grassy, and astringent notes are retained Difficulty in processing pulses with higher fat content, poor dispersibility in an airflow (e.g., chickpeas and lupins)

**Table 4 foods-14-01151-t004:** Functional properties of pulse protein concentrates and isolates produced using different fractionation methods.

Functionality	Pulse Protein	Fractionation	Results	Reference
Solubility	Faba bean (I)	WF and DF	32%; 85%, resp.	[126]
	Pea and faba bean (C)	DF	19–25%; 14–29%, resp.	[99]
	Pea (I)	WF	53.9–70.3%	[127]
	Soy (I)	WF	21%	[128]
Water-holding capacity	Lentil, mung bean, and yellow pea (I)	WF	2.9 g/g; 1.5 g/g; 2.6 g/g, resp.	[129]
	Chickpea (I)	WF	2.6 g/g	[130]
	Chickpea (I)	WF	2.3–3.5 g/g	[131]
	Mung bean, yellow pea, and cowpea (C)	DF	2.1 g/g; 1.5 g/g; 2.1 g/g resp.	[132]
	Pea and faba bean (C)	DF	0.93–0.97 g/g; 0.58–0.63 g/g	[99]
	Pea (I)	WF	1.88–2.37 g/g	[127]
	Soy and mung bean (I)	WF	3.5 g/g; 2.9 g/g, resp.	[128]
Oil-holding	Lentil, mung bean, and yellow pea (I)	WF	1.4–1.6 g/g, 1.6 g/g; 1.5 g/g, resp.	[129]
capacity	Faba, kidney bean, cowpea and field pea (I)	WF	2.3 g/g; 4.7–6.9 g/g, 1.4–2.0 g/g; 5.5–7.2 g/g, resp.	[133]
	Pea and faba bean (C)	DF	1.1 g/g; 1.1 g/g, resp.	[99]
	Pea (I)	WF	1.1–1.4 g/g	[127]
	Soy and mung bean (I)	WF	1.0 g/g; 1.1 g/g, resp.	[128]
Foaming properties	Chickpea (I)	WF	59.2–66.6%	[134]
	Faba bean (C) and faba bean (I)	DF	30–60%; 82–95%, resp.	[126]
	Pea and faba bean (I)	AC	54.8–55.5%, 26.5–41.9%, resp.	[99]
Gelation	Faba bean (I)	WF	140 g/kg (* LGC)	[135]
	Chickpea (I)	WF	140 g/L (* LGC)	[136]
	Kidney bean and field pea (I)	WF	87.4 °C; 84 °C, resp.	[133]
	Faba bean (C)	DF	65 °C	[137]

* Least gelling concentration (LGC), isolate (I), concentrate (C), dry fractionation air classification (DF), and wet fractionation (WF), respectively (resp.).

**Table 5 foods-14-01151-t005:** Pulse protein concentrate and isolate uses in the food industry.

Food Product	Pulse Protein Type	Outcomes	Reference
** *Bakery products* **			
Crackers	Faba bean (C) (I)	Improved nutritional and functional properties (lower starch and fat content)	[162]
Donuts	Lentils (C)	Increased cooking characteristics, delayed the loss of moisture, and decreased the rate of hardening	[115]
Cookies	Pea (I)	Increased water absorption capacity and rheological properties, but it does not improve spread ratio or hardness	[99]
Muffins	Lentils (C)	Improved overall score	[116]
Wheat bread	Pea (I)	Improved rheological, structural, and nutritional parameters	[163]
** *Pasta* **			
Gluten-free pasta	Pea and faba bean (I)	An overall change in colour, hardness, water uptake, and cooking losses	[164]
High protein pasta	Pea (I)	Improved chemical score, amino acid digestibility, colour, and cooking quality	[165]
** *Alternative dairy products* **			
Cheese	Chickpea (I)	Stretchability, meltability, texture, rheological, thermal behaviour, and microstructure were improved	[166]
Milk	Sweet lupin and chickpea (C)	Improved protein content	[167]
Yoghurt	Lupin (I)	Higher viscosity and with more firmness	[168]
** *Meat analogues* **			
	Pea (C) (I)	Lower hardness, high oil absorption, neutral sensory profiles	[53]
	Faba bean(C) (I)	High-moisture and fibrous structures	[169]
	Faba bean (C) (I)	Fibrous structures	[170]

Isolate (I), Concentrate (C).

## Data Availability

No data were used for the research described in the article.

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
