# Peer review of "Pulse Proteins: Processing, Nutrition, and Functionality in Foods"

_foods, 2025, doi:10.3390/foods14071151_

Round 1

Reviewer 1 Report

Comments and Suggestions for Authors

After reading the manuscript, I realized that the manuscript showed in some parts the scientific rigour wanted, but in other parts I have missed it.

The authors have presented critical evaluation only in some paragraphs.

The references are not exactly current, besides  rationale and objective are not clear.

Thats why I have written some suggestions below in an attempt to improve the paper.

The paper is too long.

This version does not align with the journal's guidelines and will need further adjustments. Please ensure that all requirements are met in the next revision.

You double-spaced lines, you didn't follow the journal's template, you didn't quote the authors with numbers.
More than 30 pages of references. It really too much...  Reorganize the references as they appear in the text and not in alphabetical order, please.

L.63 - "semi vegetarian"- please, check if flexitarian would not be better.

L.69- I think it's very important to bring up this difference, but I ask you to review it because some authors contradict themselves. "are distinguished from leguminous oil seeds (e.g., soybeans and groundnuts) as they contain much lower fat content" 

L.72- 80- I don't consider these lines to be related to the aim of the paper.

L.92- Even if it's a review paper, the rationale and objectives of the study should be clarified at the end of the Introduction.

L.95-98 -Some you have only given the genus, others the species.
Please, standardize.

L.105- Thinking of your title "Pulse proteins: Processing, Nutrition, Functionality, and Potential Uses in Functional Foods"  - Do you really think the specificity about Australia is appropriate? I'm sorry, but I don't think it's helpful.

L.172 - It was already stated on  L. 31 and L. 84. Very repetitive.

L.174- Table 5 not table 4, i guess.

L.208- "There are other  factors that can influence the quality of pulse proteins,"  - Which are ?

L.217/ 218 - "Legumins generally have a higher amount of  the sulfur-containing amino acids, methionine and cysteine" - Isn't the idea repetitive? I've read similar one.

L.239- What about protein ?

L.246- I missed a more in-depth approach to the fibers in pulses.

L.251- "between" or among ?

L.264-  Is this part really here?
Because afterwards you will have anti-nutritional factors in another section. "Protein inhibitors of the serine proteases trypsin and chymotrypsin are common in 
pulses (Aspri et al., 2023; Campos-Vega et al., 2010). Lectins or haemagglutinins found in 
pulses can inhibit the growth of animals by reducing the digestibility and biological value of 
dietary proteins (Wang et al., 2016). Pulses also have high phytate content"

L.269-  "as well as acting as an anticarcinogen in human diets" - Beware! We cannot promise an anticarcinogenic effect. We can mention that there is the possibility of secondary benefits beyond nutrition.

L.274- - "between" or among ?

L.326- What about odor ?

L.357- " Soaking" The reference you quoted Spencer, 1988- There are some authors who differ on whether soaking alone efficiently eliminates anti-nutrients. Soaking without cooking? or with cooking. Please, provide more details. This is a controversial and relevant topic.

L.417-We were discussing processing (“cooking, autoclaving, fermentation, and extrusion”), then in this section it shifts to post-harvest. It needs to be revised, a new place for this paragraph.

L.485- 678 - You focus too much on processing. Consider the title of the paper.

Pulses proteins : Potential Uses in Functional Foods - review all the sections of your paper if they are really covered.

L.1060- "sensorial" or sensory ?

L.1084 - Authors ?

L.1090- Authors,
What is the aim of the study?
You've addressed many different topics in the paper.

This makes it confusing to read.

At various times you have detailed technological processes. Definitely not the point.
Focus on the title, it seems to me the most appropriate.

L.1186 - Conclusion needs to be reviewed after the objective.

L. 2212 - Make it clearer by using the scientific name

L.2286- Since it's already 2025... this table deserves an update

Author Response

Reviewers' Comments to the Author:

Reviewer: 1

L.63 - "semi vegetarian"- please, check if flexitarian would not be better.

Changes have been made accordingly.

L.69- I think it's very important to bring up this difference, but I ask you to review it because some authors contradict themselves. "Are distinguished from leguminous oil seeds (e.g., soybeans and groundnuts) as they contain much lower fat content" 

We thank the reviewer for pointing this out. We have checked it and yes, the Food and Agriculture Organization (FAO) defines pulses as legumes with low fat content and dry, edible seeds. Pulses are distinguished from leguminous oil seeds, like soybeans and groundnuts, by their lower fat content. 

L.72- 80- I don't consider these lines to be related to the aim of the paper.

We agree with the reviewer and L72-80 has been deleted.

L.92- Even if it's a review paper, the rationale and objectives of the study should be clarified at the end of the Introduction.

We agree with the reviewer and aims have now been added.

L.95-98 -Some you have only given the genus, others the species.
Please, standardize.

Changes have been made accordingly.

L.105- Thinking of your title "Pulse proteins: Processing, Nutrition, Functionality, and Potential Uses in Functional Foods"  - Do you really think the specificity about Australia is appropriate? I'm sorry, but I don't think it's helpful.

We agree with the reviewer, and reference to Australia has been deleted.

L.172 - It was already stated on  L. 31 and L. 84. Very repetitive.

We agree with the reviewer, and it has been deleted.

L.174- Table 5 not table 4, i guess.

Number of this table has been corrected.

L.208- "There are other factors that can influence the quality of pulse proteins,"  - Which are ?

The sentence has been deleted because we explained throughout the manuscript how the quality of pulses can be changed by diverse factors.

L.217/ 218 - "Legumins generally have a higher amount of  the sulfur-containing amino acids, methionine and cysteine" - Isn't the idea repetitive? I've read similar one.

We thank the reviewer for pointing this out. We have deleted the sentence.

L.239- What about protein?

Protein content has been discussed in L106-107.

L.246- I missed a more in-depth approach to the fibers in pulses.

We agree with the reviewer, and this section has been added.

L.251- "between" or among?

Changes have been made accordingly.

L.264-  Is this part really here? Because afterwards you will have anti-nutritional factors in another section. "Protein inhibitors of the serine proteases trypsin and chymotrypsin are common in pulses (Aspri et al., 2023; Campos-Vega et al., 2010). Lectins or haemagglutinins found inpulses can inhibit the growth of animals by reducing the digestibility and biological value of dietary proteins (Wang et al., 2016). Pulses also have high phytate content"

We agree with the reviewer, and it has been moved to the antinutritional section, in which phytate content has been discussed.

L.269-  "as well as acting as an anticarcinogen in human diets" - Beware! We cannot promise an anticarcinogenic effect. We can mention that there is the possibility of secondary benefits beyond.

We thank the reviewer for pointing this out and apologize for the mistake. The sentence has been revised accordingly.

L.274- - "between" or among?

Changes have been made accordingly.

L.326- What about Odor?

We thank the reviewer for pointing this out, but we were focused on off-flavours because it can be a significant problem to the sensory qualities of pulse-based foods, consumer acceptance, and the overall quality of pulse-based products. While odours can make a food unappealing, the sensory properties can be improved by processing or cooking the food.

L.357- " Soaking" The reference you quoted Spencer, 1988- There are some authors who differ on whether soaking alone efficiently eliminates anti-nutrients. Soaking without cooking? or with cooking. Please, provide more details. This is a controversial and relevant topic.

We thank the reviewer and agree with the suggestion. We have updated the reference.

L.417-We were discussing processing (“cooking, autoclaving, fermentation, and extrusion”), then in this section it shifts to post-harvest. It needs to be revised, a new place for this paragraph.

We thank the reviewer and agree with the suggestion. The paragraph and figure have been relocated to the 'Worldwide Pulse Production' section for better clarity and context.

L.485- 678 - You focus too much on processing. Consider the title of the paper.Pulses proteins: Potential Uses in Functional Foods - review all the sections of your paper if they are really covered.

We thank the reviewer for this valuable feedback. However, we believe it is important to provide an overview of techniques for protein extraction from pulses. We have revised and shortened this section to make it more concise and comprehensive.

L.1060- "sensorial" or sensory?

Changes have been made accordingly.

L.1084 - Authors?

Changes have been made accordingly.

L.1090- Authors,
What is the aim of the study?You've addressed many different topics in the paper. This makes it confusing to read. At various times you have detailed technological processes. Definitely not the point. Focus on the title, it seems to me the most appropriate.

We agree with the reviewer, and this section has been deleted.

L.1186 - Conclusion needs to be reviewed after the objective.

Changes have been made accordingly.

  1. 2212 - Make it clearer by using the scientific name

Changes have been made accordingly.

L.2286- Since it's already 2025... this table deserves an update

The Food and Agriculture Organization (FAO) has not yet released comprehensive global pulse production statistics for 2023 and 2024. The most recent data available is up to 2022. Due to this the table has been updated to 2022.

Reviewer 2 Report

Comments and Suggestions for Authors

This article primarily explores the nutritional significance of pulse crops including lupin, chickpeas, faba beans, peas, lentils, and green beans, underscoring their relevance in agriculture and food processing. Key topics covered include: the nutritional attributes of pulse crops, the quality and functionality of their protein, the health benefits they offer, their potential contribution to global food security, and the utilization of pulse protein in food manufacturing. Additionally, the paper examines both established and emerging technologies pertinent to this field. This review aims to serve as a comprehensive resource for researchers, food technologists, and food manufacturers, aiding in their understanding of the diverse applications and emerging trends related to pulses. The review is thoroughly composed and comprehensive. The recommendations are deemed acceptable following minor revisions. The specific modification suggestions are outlined as follows:

  1. To elucidate the role of Pulse proteins in disease prevention and treatment, it is imperative to provide a concise overview of relevant molecular mechanisms and offer an initial synthesis.
  2. One pertinent review concerning the neuroprotective effects of legumes was inadvertently omitted by the authors and should be cited along with a brief discussion (Ye X-S, Tian W-J, Wang G-H, et al. The food and medicine homologous Chinese Medicine from Leguminosae species: A comprehensive review on bioactive constituents with neuroprotective effects on nervous system. Food & Medicine Homology, 2025, 2(2): 9420033. https://doi.org/10.26599/FMH.2025.9420033; Rajpurohit B, Li Y. Overview on pulse proteins for future foods: Ingredient development and novel applications. Journal of Future Foods, 2023, 3(4): 340-356. https://doi.org/10.1016/j.jfutfo.2023.03.005).
  3. The author should incorporate an in-depth discussion regarding the correlation between Pulse proteins and tumor diseases. This topic is well-documented in existing literature and holds significant importance for a comprehensive understanding of the subject (M.-Y. Li, A. Gu, J. Li, N. Tang, M. Matin, Y. Yang, G. Zengin, A. G. Atanasov. Exploring food and medicine homology: potential implications for cancer treatment innovations. Acta Materia Medica 2025, 4, 200-206. DOI: 10.15212/AMM-2025-0003).
  4. The authors should comprehensively examine and discuss the impact of various processing methods on the nutritional composition and associated functional properties.
  5. How do you think Pulse proteins will affect the future food?

Author Response

Reviewers' Comments to the Author:

Reviewer: 2

  1. To elucidate the role of Pulse proteins in disease prevention and treatment, it is imperative to provide a concise overview of relevant molecular mechanisms and offer an initial synthesis.
  2. One pertinent review concerning the neuroprotective effects of legumes was inadvertently omitted by the authors and should be cited along with a brief discussion (Ye X-S, Tian W-J, Wang G-H, et al. The food and medicine homologous Chinese Medicine from Leguminosae species: A comprehensive review on bioactive constituents with neuroprotective effects on nervous system. Food & Medicine Homology, 2025, 2(2): 9420033. https://doi.org/10.26599/FMH.2025.9420033; Rajpurohit B, Li Y. Overview on pulse proteins for future foods: Ingredient development and novel applications. Journal of Future Foods, 2023, 3(4): 340-356. https://doi.org/10.1016/j.jfutfo.2023.03.005).
  3. The author should incorporate an in-depth discussion regarding the correlation between Pulse proteins and tumor diseases. This topic is well-documented in existing literature and holds significant importance for a comprehensive understanding of the subject (M.-Y. Li, A. Gu, J. Li, N. Tang, M. Matin, Y. Yang, G. Zengin, A. G. Atanasov. Exploring food and medicine homology: potential implications for cancer treatment innovations. Acta Materia Medica 2025, 4, 200-206. DOI: 10.15212/AMM-2025-0003).

We thank the reviewer for these suggestions. However, the primary aim of this review was to provide a comprehensive resource on pulses, with a focus on protein quality, functionality, and the diverse applications of pulse protein ingredients. While health benefits were briefly mentioned in some sections, the review was not intended to offer a detailed overview of health-related aspects or include relevant molecular mechanisms (Point 1, 2 and 3).

  1. The authors should comprehensively examine and discuss the impact of various processing methods on the nutritional composition and associated functional properties.

The impact of processing methods on the nutritional composition and associated functional properties has been discussed throughout the review.

  1. How do you think Pulse proteins will affect the future food?

We thank the reviewer for these positive comments. Their suggestions have been incorporated into the conclusions.

Round 2

Reviewer 1 Report

Comments and Suggestions for Authors

After another evaluation of the manuscript, I see a great improvement in the quality of the paper. The authors have accepted some of my requests, but there is still room for improvement.

L.2 - "Functional Foods " - Are you sure that's what you mean?

L.43- The rationale for the study is not yet well established, and I mean that it does not cover the objectives and title.

L.55- Off flavor ?  Extraction ? Processing ?Do you really believe that the objective covers all the subjects in the paper?

L.138- Paragraph

Review the entire paper Foods rules for quoting authors, please. I pointed out only some of them below.

L.155-  Siah et al. (2014)  - Check Foods rules for quoting authors.

L.164- Johnson et al. (2020) -  Check Foods rules for quoting authors.

L.212- Amarakoon et al. (2012) -  Check Foods rules for quoting authors.

L.232- Godrich et al., (2023) -  Check Foods rules for quoting authors.

L.537- Xing et al. (2021) 

L.649- Osen et al. (2014)

L.656- Kantanen et. al. (2022) 

L.658 De Angelis et al. (2020)

L.665- Zhu et al. (2021) -  Check Foods rules for quoting authors.

L.673 Penchalaraju et al. (2024) - Check Foods rules for quoting authors.

There are still many specific references on Australia. Please check for appropriateness.There are still scientific names without italics, for example in the references.

Author Response

Reviewers' Comments to the Author:

-L.2 - "Functional Foods " - Are you sure that's what you mean?

We agree with the reviewer, and it has been changed.

-L.43- The rationale for the study is not yet well established, and I mean that it does not cover the objectives and title.

We agree with the reviewer, and it has been added.

-L.55- Off flavor ?  Extraction ? Processing ?Do you really believe that the objective covers all the subjects in the paper?

We agree with the reviewer, and it has been changed.

--L.138- Paragraph

We agree with the reviewer, and it has been changed.

Review the entire paper Foods rules for quoting authors, please.

Changes have been made accordingly.

-There are still many specific references on Australia. Please check for appropriateness.

We have removed most of the references specific to Australia as you suggested in the previous review, but as all authors are researchers working in the pulse field and based in Australia, we felt it would be appropriate to keep some local references, while also incorporating global references to highlight advancements worldwide. This balance ensures that the review acknowledges the local contributions to pulse protein research while remaining relevant to the broader global context.

-There are still scientific names without italics, for example in the references

Changes have been made accordingly.
